# AReLU: Attention-based Rectified Linear Unit

## Abstract

Element-wise activation functions play a critical role in deep neural networks via affecting the expressivity power and the learning dynamics. Learning-based activation functions have recently gained increasing attention and success. We propose a new perspective of learnable activation function through formulating them with *element-wise attention mechanism*. In each network layer, we devise an attention module which learns an element-wise, sign-based attention map for the pre-activation feature map. The attention map scales an element based on its sign. Adding the attention module with a rectified linear unit (ReLU) results in an amplification of positive elements and a suppression of negative ones, both with learned, data-adaptive parameters. We coin the resulting activation function Attention-based Rectified Linear Unit (AReLU). The attention module essentially learns an element-wise residue of the activated part of the input, as ReLU can be viewed as an identity transformation. This makes the network training more resistant to gradient vanishing. The learned attentive activation leads to well-focused activation of relevant regions of a feature map. Through extensive evaluations, we show that AReLU significantly boosts the performance of most mainstream network architectures with only two extra learnable parameters per layer introduced. Notably, AReLU facilitates fast network training under small learning rates, which makes it especially suited in the case of transfer learning and meta learning.

## 1  Introduction

Activation functions, introducing nonlinearities to artificial neural networks, is essential to networks' expressivity power and learning dynamics. Designing activation functions that facilitate fast training of accurate deep neural networks is an active area of research (Maas et al., 2013; Goodfellow et al., 2013; Xu et al., 2015a; Clevert et al., 2015; Hendrycks & Gimpel, 2016; Klambauer et al., 2017; Barron, 2017; Ramachandran et al., 2017). Aside from the large body of hand-designed functions, learning-based approaches recently gain more attention and success (Agostinelli et al., 2014; He et al., 2015; Manessi & Rozza, 2018; Molina et al., 2019; Goyal et al., 2019). The existing learnable activation functions are motivated either by relaxing/parameterizing a non-learnable activation function (e.g. Rectified Linear Units (ReLU) (Nair & Hinton, 2010)) with learnable parameters (He et al., 2015), or by seeking for a data-driven combination of a pool of pre-defined activation functions (Manessi & Rozza, 2018). Existing learning-based methods make activation functions data-adaptive through introducing degrees of freedom and/or enlarging the hypothesis space explored.

In this work, we propose a new perspective of learnable activation functions through formulating them with *element-wise attention mechanism*. A straightforward motivation of this is a straightforward observation that both activation functions and element-wise attention functions are applied as a network module of *element-wise multiplication*. More intriguingly, learning element-wise activation functions in a neural network can intuitively be viewed as task-oriented attention mechanism (Chorowski et al., 2015; Xu et al., 2015b), i.e., *learning where (which element in the input feature map) to attend (activate) given an end task to fulfill*. This motivates an arguably more interpretable formulation of *attentive activation functions*.

Attention mechanism has been a cornerstone in deep learning. It directs the network to learn which part of the input is more relevant or contributes more to the output. There have been many variants of attention modules with plentiful successful applications. In natural language processing, vector-wise attention is developed to model the long-range dependencies in a sequence of word vectors (Luong et al., 2015; Vaswani et al., 2017). Many computer vision tasks utilize pixel-wise or channel-wise attention modules for more expressive and invariant representation learning (Xu et al., 2015b; Chen

et al., 2017). Element-wise attention (Bochkovskiy et al., 2020) is the most fine-grained where each element of a feature volume can receive different amount of attention. Consequently, it attains high expressivity with neuron-level degrees of freedom.

Inspired by that, we devise for each layer of a network an element-wise attention module which learns a sign-based attention map for the pre-activation feature map. The attention map scales an element based on its sign. Through adding the attention and a ReLU module, we obtain Attention-based Rectified Linear Unit (AReLU) which amplifies positive elements and suppresses negative ones, both with learned, data-adaptive parameters. The attention module essentially learns an element-wise residue for the activated elements with respect to the ReLU since the latter can be viewed as an identity transformation. This helps ameliorate the gradient vanishing issue effectively. Through extensive experiments on several public benchmarks, we show that AReLU significantly boosts the performance of most mainstream network architectures with only two extra learnable parameters per layer introduced. Moreover, AReLU enables fast learning under small learning rates, making it especially suited for transfer learning. We also demonstrate with feature map visualization that the learned attentive activation achieves well-focused, task-oriented activation of relevant regions.

## 2 RELATED WORK

**Non-learnable activation functions**  Sigmoid is a non-linear, saturated activation function used mostly in the output layers of a deep learning model. However, it suffers from the exploding/vanishing gradient problem. As a remedy, the rectified linear unit (ReLU) (Nair & Hinton, 2010) has been the most widely used activation function for deep learning models with the state-of-the-art performance in many applications. Many variants of ReLU have been proposed to further improve its performance on different tasks LReLU (Maas et al., 2013), ReLU6 (Krizhevsky & Hinton, 2010), RReLU (Xu et al., 2015a). Besides that, some specified activation functions also have been designed for different usages, such as CELU (Barron, 2017), ELU (Clevert et al., 2015), GELU (Hendrycks & Gimpel, 2016), Maxout (Goodfellow et al., 2013), SELU (Klambauer et al., 2017), (Softplus) (Glorot et al., 2011), Swish (Ramachandran et al., 2017).

**Learnable activation functions**  Recently, learnable activation functions have drawn more attentions. PReLU (He et al., 2015), as a variants of ReLU, improves model fitting with little extra computational cost and overfitting risk. Recently, PAU (Molina et al., 2019) is proposed to not only approximate common activation functions but also learn new ones while providing compact representations with few learnable parameters. Several other learnable activation functions such as APL (Agostinelli et al., 2014), Comb (Manessi & Rozza, 2018), SLAF (Goyal et al., 2019) also achieve promising performance under different tasks.

**Attention Mechanism**  Vector-Wise Attention Mechanism (VWAM) has been widely applied in Natural Language Processing (NLP) tasks (Xu et al., 2015c; Luong et al., 2015; Bahdanau et al., 2014; Vaswani et al., 2017; Ahmed et al., 2017). VWAM learns which vector among a sequence of word vectors is the most relevant to the task in hand. Channel-Wise Attention Mechanism (CWAM) can be regarded as an extension of VWAM from NLP to Vision tasks (Tang et al., 2019b; 2020; Kim et al., 2019). It learns to assign each channel an attentional value. Pixel-Wise Attention Mechanism (PWAM) is also widely used in vision (Tang et al., 2019c;a). Element-Wise Attention Mechanism (EWAM) assigns different values to each element without any spatial/channel constraint. The recently proposed YOLOv4 (Bochkovskiy et al., 2020) is the first work that introduces EWAM implemented by a convolutional layer and sigmoid function. It achieves the state-of-the-art performance on object detection. We introduce a new kind of EWAM for learnable activation function.

## 3 METHOD

We start by describing attention mechanism and then introduce element-wise sign-based attention mechanism based on which AReLU is defined. The optimization of AReLU then follows.

### 3.1 ATTENTION MECHANISM

Let us denote $V = \{v_i\} \in \mathbb{R}^{D_v^1 \times D_v^2 \times \cdots}$ a tensor representing input data or feature volume. Function $\Phi$, parameterized by $\Theta = \{\theta_i\}$, is used to compute an attention map $S = \{s_i\} \in \mathbb{R}^{D_v^{\theta(1)} \times D_v^{\theta(2)} \times \cdots}$

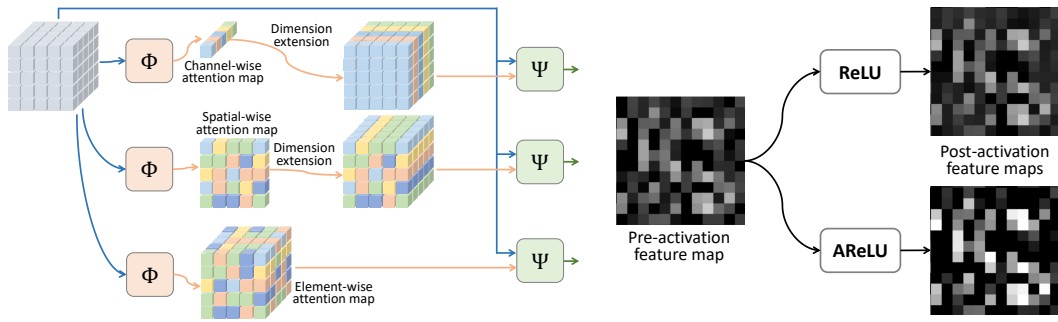

Figure 1: Left: An illustration of attention mechanisms with attention map at different granularities. Right: Visualization of pre-activation and post-activation feature maps obtained with ReLU and AReLU on a testing image of the handwritten digit dataset MNIST LeCun et al. (1998).

over a subspace of $V$ (let $\theta(\cdot)$ denote a correspondence function for the indices of dimension):

$$s_i = \Phi(v_i, \Theta). \tag{1}$$

$\Phi$ can be implemented by a neural network with $\Theta$ being its learnable parameters.

We can modulate the input $V$ with the attention map $S$ using a function $\Psi$, obtaining the output $U = \{u_i\} \in \mathbb{R}^{D_v^1 \times D_v^2 \times \cdots}$:

$$u_i = \Psi(v_i, s_i). \tag{2}$$

$\Psi$ is an element-wise multiplication. In order to perform element-wise multiplication, one needs to first extend $S$ to the full dimension of $V$. We next review various attention mechanisms with attention map at different granularities. Figure 1(left) gives an illustration of various attention mechanisms.

**Vector-wise Attention Mechanism** In NLP, attention maps are usually computed over different word vectors. In this case, $V = \{v_i\} \in \mathbb{R}^{N \times D}$ represents a sequence of $N$ feature vectors with dimension $D$. $S = \{s_i\} \in \mathbb{R}^N$ is a sequence of attention values for the corresponding vectors.

**Channel-wise Attention Mechanism** In computer vision, a feature volume $V = \{v_i\} \in \mathbb{R}^{W \times H \times C}$ has a spatial dimension of $W \times H$ and a channel dimension of $C$. $S = \{s_i\} \in \mathbb{R}^C$ is an attention map over the $C$ channels. All elements in each channel share the same attention value.

**Spatial-wise Attention Mechanism** Considering again $V = \{v_i\} \in \mathbb{R}^{W \times H \times C}$ with a spatial dimension of $W \times H$. $S = \{s_i\} \in \mathbb{R}^{W \times H}$ is an attention map over the spatial dimension. All channels of a given spatial location share the same attention value.

**Element-wise Attention Mechanism** Given a feature volume $V = \{v_i\} \in \mathbb{R}^{W \times H \times C}$ containing $W \times H \times C$ elements, we compute an attention map over the whole volume (all elements), i.e., $S = \{s_i\} \in \mathbb{R}^{W \times H \times C}$, so that each element has an independent attention value.

### 3.2 Element-wise Sign-based Attention (ELSA)

We propose, ELSA, a new kind of element-wise attention mechanism which is used to define our attention-based activation. Considering a feature volume $V = \{v_i\} \in \mathbb{R}^{W \times H \times C}$, we compute an element-wise attention map $S = \{s_i\} \in \mathbb{R}^{W \times H \times C}$:

$$s_i = \Phi(v_i, \Theta) = \begin{cases} C(\alpha), & v_i < 0 \\ \sigma(\beta), & v_i \geq 0 \end{cases} \tag{3}$$

where $\Theta = \{\alpha, \beta\} \in \mathbb{R}^2$ is learnable parameters. $C(\cdot)$ clamps the input variable into $[0.01, 0.99]$. $\sigma$ is the sigmoid function. The modulation function $\Psi$ is defined as:

$$u_i = \Psi(v_i, s_i) = s_i v_i. \tag{4}$$

In ELSA, positive and negative elements receive different amount of attention determined by the two parameters $\alpha$ and $\beta$, respectively. Therefore, it can also be regarded as sign-wise attention mechanism. With only two learnable parameters, ELSA is light-weight and easy to learn.

### 3.3 AReLU: Attention-based Rectified Linear Units

We represent the function $\Phi$ in ELSA with a network layer with learnable parameters $\alpha$ and $\beta$:

$$\mathcal{L}(x_i, \alpha, \beta) = \begin{cases} C(\alpha)x_i, & x_i < 0 \\ \sigma(\beta)x_i, & x_i \geq 0 \end{cases} \tag{5}$$

where $X = \{x_i\}$ is the input of the current layer. In constructing an activation function with ELSA, we combine it with the standard Rectified Linear Units

$$\mathcal{R}(x_i) = \begin{cases} 0, & x_i < 0 \\ x_i, & x_i \geq 0 \end{cases} \tag{6}$$

Adding them together leads to a learnable activation function:

$$\mathcal{F}(x_i, \alpha, \beta) = \mathcal{R}(x_i) + \mathcal{L}(x_i, \alpha, \beta) = \begin{cases} C(\alpha)x_i, & x_i < 0 \\ (1 + \sigma(\beta))x_i, & x_i \geq 0 \end{cases} \tag{7}$$

This combination amplifies positive elements and suppresses negative ones based on the learned scaling parameters $\beta$ and $\alpha$, respectively. Thus, ELSA learns an element-wise residue for the activated elements w.r.t. ReLU which is an identity transformation, which helps ameliorate gradient vanishing.

### 3.4 The Optimization of AReLU

AReLU can be trained using back-propagation jointly with all other network layers. The update formulation of $\alpha$ and $\beta$ can be derived with the chain rule. Specifically, the gradient of $\alpha$ is:

$$\frac{\partial \mathcal{E}}{\partial \alpha} = \frac{\partial \mathcal{E}}{\partial \mathcal{F}(x_i, \alpha, \beta)} \frac{\partial \mathcal{F}(x_i, \alpha, \beta))}{\partial \alpha} \tag{8}$$

where $\mathcal{E}$ is the error function to be minimized. The term $\frac{\partial \mathcal{E}}{\partial \mathcal{F}(x_i, \alpha, \beta)}$ is the gradient propagated from the deeper layer. The gradient of the activation of $X$ with respect to $\alpha$ is given by:

$$\frac{\partial \mathcal{F}(X, \alpha, \beta)}{\partial \alpha} = \sum_{x_i < 0} x_i \tag{9}$$

Here, the derivative of the clamp function $C(\cdot)$ is handled simply by detaching the gradient back-propagation when $\alpha < 0.01$ or $\alpha > 0.99$.

The gradient of the activation of $X$ with respect to $\beta$ is:

$$\frac{\partial \mathcal{F}(X, \alpha, \beta)}{\partial \beta} = \sum_{x_i \geq 0} \sigma(\beta)(1 - \sigma(\beta))x_i \tag{10}$$

The gradient of the activation with respect to input $x_i$ by:

$$\frac{\partial \mathcal{F}(x_i, \alpha, \beta)}{\partial x_i} = \begin{cases} \alpha, & x_i < 0 \\ 1 + \sigma(\beta), & x_i \geq 0 \end{cases} \tag{11}$$

It can be found that AReLU amplifies the gradients propagated from the downstream when the input is activated since $1 + \sigma(\beta) > 1$; it suppresses the gradients otherwise. On the contrary, there is no such amplification effect in the standard ReLU and its variants (e.g., PReLu (He et al., 2015)) — only suppression is available. The ability to amplify the gradients over the activated input helps avoiding gradient vanishing, and thus speeds up the training convergence of the model (see Figure 3). Moreover, the amplification factor is learned to dynamically adapt to the input and is confined with the sigmoid function. This makes the activation more data-adaptive and stable (see Figure 1(right) for a visual comparison of post-activation feature maps by AReLU and ReLU). The suppression part is similar to PReLu which learns the suppression factor for ameliorating zero gradients.

AReLU introduces a very small number of extra parameters which is $2L$ for an $L$-layer network. The computational complexity due to AReLU is negligible for both forward and backward propagation.

Note that the gradients of $\alpha$ and $\beta$ depend on the entire feature volume $X$. This means that ELSA can be regarded as a global attention mechanism: Although the attention map is computed in an element-wise manner, the parameters are learned globally accounting for the impact of the full feature volume. This makes our AReLU more data-adaptive and hence the whole network more expressive.

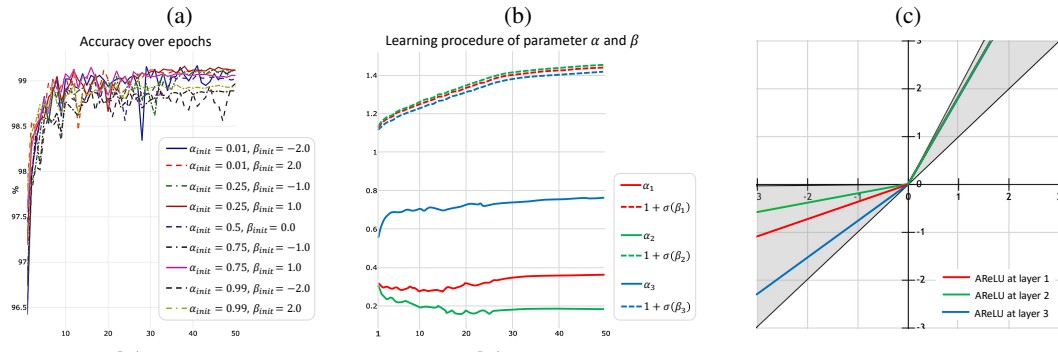

Figure 2: (a): Plot of accuracy over epochs for networks trained with different initialization of $\alpha$ and $\beta$. A larger initial $\beta$ leads to faster convergence and higher accuracy is obtained when $\alpha$ is initialized to $0.25$ or $0.75$. (b): The learning procedure of $\alpha$ and $\beta$ which are initialized to $0.25$ and $1.0$, respectively. (c): The learned final AReLU's for the three convolutional layers of the MNIST-Conv network. The shaded region gives the range of AReLU curves.

## 4 EXPERIMENTS

We first study the robustness of AReLU in terms of parameter initialization. We then evaluate convergence of network training with different activation functions on two standard classification benchmarks (MNIST (LeCun et al., 1998)) and CIFAR100 (Krizhevsky et al., 2009). We compare AReLU with 18 different activation functions including 13 non-learnable ones and 5 learnable ones; see the list in Table 1. The number of learnable parameters for each learnable activation function are also given in the table. In the end, we also demonstrate the advantages of AReLU in transfer learning. *Please refer to supplemental material for more results and experiments details.*

### 4.1 INITIALIZATION OF LEARNABLE PARAMETER $\alpha$ AND $\beta$

For evaluation purpose, we design a neural network (MNIST-Conv) with three convolutional layers each followed by a max-pooling layer and an AReLU, and finally a fully connected layer followed by a softmax layer. *Details of this network can be found in the supplemental material.* The experiment on parameter initialization is conducted with MNIST-Conv over the MNIST dataset. As shown in Figure 2(a), AReLU is insensitive to the initialization of $\alpha$ and $\beta$. Different initial values result in close convergence rate and classification accuracy. Generally, a large initial value of $\beta$ can speed up the convergence. Figure 2(b) shows the learning procedure of the two parameters and (c) plots the learned final AReLU's for the three convolutional layers. In the following experiments, we initialize $\alpha = 0.9$ and $\beta = 2.0$ by default.

### 4.2 CONVERGENCE ON MNIST

On the MNIST dataset, we evaluate MNIST-Conv implemented with different activation functions and trained with the ADAM or SGD optimizer. The activation function is placed after each max-pooling layers. We compare AReLU with both learnable and non-learnable activation functions under different learning rates of $1 \times 10^{-2}$, $1 \times 10^{-3}$, $1 \times 10^{-4}$, and $1 \times 10^{-5}$. To compare the convergence speed of different activation functions, we report the accuracy after the first epoch, again taking the mean over five times training; see Table 1. In the table, we report the improvement of AReLU over the best among other non-learnable and learnable methods. In Figure 3, we plot the mean accuracy over increasing number of training epochs.

As shown in Table 1, AReLU outperforms most existing non-learnable and learnable activation functions in terms of convergence speed and final classification accuracy on MNIST. A note-worthy phenomenon is that AReLU can achieve a more effective training with a small learning rate (see the significant improvement when the learning rate is $1 \times 10^{-4}$ or $1 \times 10^{-5}$) than the alternatives. This can also be observed from Figure 3. Generally, smaller learning rates would cause lower learning efficiency since the vanishing gradient issue is intensified in such case. AReLU can overcome this difficulty thanks to its gradient amplification effect. Efficient learning with a small learning rate is very useful in transfer learning where a pre-trained model is usually fine-tuned on a new domain/dataset

Table 1: Mean testing accuracy (%) on MNIST for five trainings of MNIST-Conv after the *first epoch* with different optimizers and learning rates. We compare AReLU with 13 non-learnable and 5 learnable activation functions. The number of parameters per activation unit are listed beside the name of the learnable activation functions. The best numbers are shown in bold text with blue color for non-learnable methods (the upper part of the table) and red for learnable ones (the lower part). At the bottom of the table, we report the improvement of AReLU over the best among other non-learnable and learnable methods, in blue and red color respectively.

| Learning Rate | $1 \times 10^{-2}$ | | $1 \times 10^{-3}$ | | $1 \times 10^{-4}$ | | $1 \times 10^{-5}$ | |
|---|---|---|---|---|---|---|---|---|
| Optimizer | Adam | SGD | Adam | SGD | Adam | SGD | Adam | SGD |
| CELU (2017) | 97.76 | 96.12 | 96.21 | 62.81 | 84.01 | 13.07 | 24.84 | 9.60 |
| ELU (2015) | 97.82 | 96.17 | 96.22 | 58.10 | **85.67** | 14.07 | 19.77 | 10.13 |
| GELU (2016) | **98.49** | 94.90 | 95.79 | 12.55 | 83.72 | 11.49 | 15.20 | **10.92** |
| LReLU (2013) | 97.80 | 95.59 | 95.86 | 35.90 | 84.08 | 10.28 | 15.41 | 10.73 |
| Maxout (2013) | 97.04 | 95.81 | 96.14 | 71.75 | 84.81 | 10.79 | 18.83 | 9.06 |
| ReLU (2010) | 97.75 | 95.02 | 95.40 | 36.01 | 84.02 | 10.68 | 15.25 | 8.73 |
| ReLU6 (2010) | 97.77 | 95.32 | 96.09 | 43.42 | 81.39 | 10.23 | 14.33 | 9.56 |
| RReLU (2015a) | 98.09 | 95.88 | 95.65 | 53.33 | 84.51 | 9.57 | 16.53 | 10.28 |
| SELU (2017) | 97.25 | **96.52** | **96.61** | **82.36** | 85.36 | **16.49** | **30.04** | 9.59 |
| Sigmoid | 47.16 | 11.04 | 83.59 | 11.35 | 11.37 | 9.92 | 10.52 | 10.10 |
| Softplus (2011) | 96.38 | 90.90 | 93.83 | 11.14 | 51.83 | 9.19 | 10.21 | 9.89 |
| Swish (2017) | 98.10 | 94.02 | 95.91 | 11.44 | 83.91 | 10.69 | 11.39 | 9.47 |
| Tanh | 96.93 | 94.22 | 96.45 | 57.70 | 79.25 | 11.73 | *27.05* | 10.31 |
| APL (2014) (2) | 97.00 | 95.71 | 94.67 | 17.81 | 76.73 | 9.39 | 13.28 | 11.83 |
| Comb (2018) (1) | **98.28** | 95.97 | 95.79 | 35.95 | 83.91 | 10.59 | 20.22 | 10.18 |
| PAU (2019) (10) | 98.17 | **97.67** | 96.73 | 40.11 | 87.08 | 10.54 | 14.49 | 11.11 |
| PReLU (2015) (1) | 98.22 | 95.72 | 95.87 | 45.73 | 85.81 | 12.08 | 14.51 | 9.88 |
| SLAF (2019) (2) | 96.30 | 97.07 | 95.32 | 83.35 | 72.67 | 14.12 | 10.04 | 11.32 |
| AReLU (2) | 98.00 | 97.30 | **97.13** | **93.13** | **90.44** | **47.78** | **38.39** | **14.25** |
| Improvement | −0.49 | +0.78 | +0.52 | +10.77 | +4.77 | +31.29 | +8.35 | +3.33 |
| Improvement | −0.28 | −0.37 | +0.40 | +9.78 | +3.36 | +33.66 | +18.17 | +2.42 |

Figure 3: Plots of mean testing accuracy (%) on MNIST for five-time trainings of MNIST-Conv over increasing training epochs. The training is conducted using SGD with small learning rates (left: $1 \times 10^{-4}$, right: $1 \times 10^{-5}$).

with a small learning rate which is difficult for most existing deep networks. Section 4.4 will demonstrate this application of AReLU.

## 4.3 CONVERGENCE ON CIFAR100

In order to better demonstrate the effect of ELSA, we regard the ReLU, without ELSA, as our baseline. For plot clarity, we choose to compare only with those most representative competitive activation functions including PAU, SELU, ReLU, LReLU (LReLU), and PReLU. *More results can be found in the supplemental material.* We evaluate the performance of AReLU with five different mainstream network architectures on CIFAR100. We use the SGD optimizer and follow the training configuration in (Pereyra et al., 2017): The learning rate is 0.1, the batch size is 64, the weight decay is $5 \times 10^{-4}$, and the momentum is 0.9.

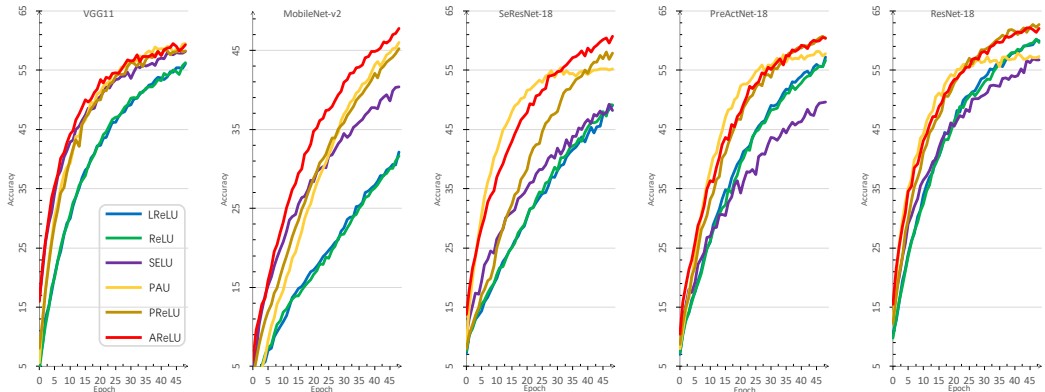

Figure 4: Plots of mean testing accuracy (%) on CIFAR100 over increasing training epochs, using different network architectures. The training is conducted using SGD with a learning rate of 0.1.

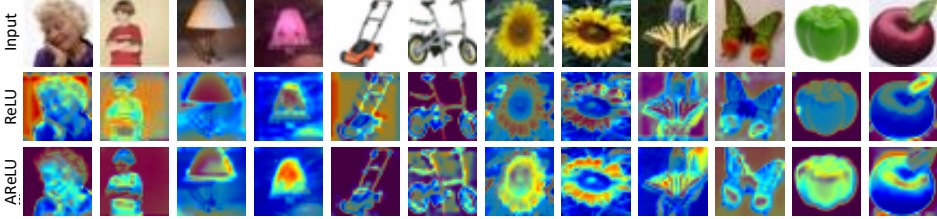

Figure 5: Grad-CAM visualization of feature maps extracted by ResNet-18 with AReLU and ReLU. The first row is the testing images of CIFAR100.

The results are plotted in Figure 4. Learnable activation functions generally have a faster convergence compared to non-learnable ones. AReLU achieves a faster convergence speed for all the five network architectures. It is worth to note that though PAU can achieve a faster convergence at the beginning in some networks such as SeResNet-18, it tends to overfit later with a fast saturation of accuracy. AReLU avoids such overfitting with smaller number of parameters than PAU (2 vs 10).

We also conduct a qualitative analysis of AReLU by visualizing the learned feature maps using Grad-CAM Selvaraju et al. (2017) using testing images of CIFAR100. Grad-CAM is a recently proposed network visualization method which utilizes gradients to depict the importance of the spatial locations in a feature map. Since gradients are computed with respect to a specific image class, Grad-CAM visualization can be regarded as a task-oriented attention map. In Figure 5, we visualize the first-layer feature map of ResNet-18. As shown in the figure, the feature maps learned with AReLU leads to semantically more meaningful activation of regions with respect to the target class. This is due to the data-adaptive, attentive ability of AReLU.

## 4.4 PERFORMANCE IN TRANSFER LEARNING

We evaluate transfer learning of MNIST-Conv with different activation functions between two datasets: MNIST and SVHN[1]. The data preprocessing for adapting the two datasets follows (Shin et al., 2017). We train three models and test them on SVHN: 1) one is trained directly on SVHN without any pretraining, 2) one trained on MNIST but not finetuned on SVHN, and 3) one pretrained on MNIST and finetuned on SVHN. In pretraining, we train MNIST-Conv using SGD with a learning rate of 0.01 for 20 epochs which is sufficient for all model variants to converge. In finetuning, we train the model on SVHN with a learning rate of $1 \times 10^{-5}$, using SGD optimizer for 100 epochs.

The testing results on SVHN are reported in Table 2 where we compare AReLU with several competitive alternatives. Without pretraining, it is hard to obtain a good accuracy on the difficult task of SVHN. Nevertheless, MNIST-Conv with AReLU performs the best among all alternatives; some activation functions even failed in learning. In the setting of transfer learning (pretrain + finetune), AReLU outperforms all other activation functions for different amount of pretraining, thanks to it high learning efficiency with small learning rates.

---

[1]http://ufldl.stanford.edu/housenumbers/

Table 2: Test accuracy (%) on SVHN by MNIST-Conv models (implemented with different activation functions) trained directly on SVHN (no pretrain), trained on MNIST but not finetuned (no finetune), as well as pretrained on MNIST and finetuned on SVHN for 5, 10 and 20 epoches. The left part of the table is non-learnable activation functions and the right learnable ones.

| Setting | ELU | GELU | Maxout | ReLU | SELU | Softplus | APL | Comb | PAU | PReLU | SLAF | AReLU |
|---|---|---|---|---|---|---|---|---|---|---|---|---|
| no pretrain | 19.59 | 19.59 | 23.01 | 19.58 | 19.58 | 19.58 | 19.58 | 19.58 | 19.58 | 19.58 | 19.58 | **24.95** |
| no finetune | 31.95 | **37.38** | 36.52 | 36.87 | 32.57 | 14.39 | 36.20 | 35.89 | 24.67 | 33.45 | 35.74 | 31.91 |
| f.t. 5 epochs | 70.08 | 69.19 | 70.18 | 69.76 | 72.81 | 65.81 | 71.73 | 69.63 | 75.24 | 66.13 | 75.91 | **76.68** |
| f.t. 10 epochs | 70.83 | 71.69 | 71.31 | 71.38 | 72.11 | 69.14 | 73.51 | 70.31 | 76.26 | 67.63 | 76.21 | **78.12** |
| f.t. 20 epochs | 71.48 | 70.34 | 73.29 | 72.06 | 71.55 | 71.01 | 72.41 | 72.55 | 74.46 | 71.99 | 74.38 | **78.48** |

Table 3: Test accuracy (%) on MNIST by MAML with MNIST-Conv models implemented with different activation functions. The performance is compared on a 5-ways-1-shots task and a 5-ways-5-shots task, respectively.

| (ways, shots) | ELU | GELU | Maxout | ReLU | SELU | Softplus | APL | Comb | PAU | PReLU | SLAF | AReLU |
|---|---|---|---|---|---|---|---|---|---|---|---|---|
| (5, 1) | 82.50 | 81.88 | 83.13 | 70.00 | 83.75 | 25.63 | 71.25 | 75.63 | 43.13 | 88.13 | 84.38 | **92.50** |
| (5, 5) | **94.30** | 69.37 | 93.12 | 78.00 | 93.50 | 22.12 | 63.00 | 88.25 | 40.12 | 91.75 | 77.00 | **94.30** |

## 4.5 Performance in Meta Learning

We evaluate the meta learning performance of MNIST-Conv with the various activation functions based on the MAML framework Finn et al. (2017). MAML is a fairly general optimization-based algorithm compatible with any model that learns through gradient descent. It aims to obtain meta-learning parameters from similar tasks and adapt the parameters to novel tasks with the same distribution using a few gradient updates. In MAML, model parameters are explicitly trained such that a small number of gradient updates over a small amount of training data from the novel task could lead to good generalization performance on that task. We expect that the fast convergence of AReLU would help MAML to adapt a model to a novel task more efficiently and with better generalization. We set the fast adaption steps as 5 and use 32 tasks for each steps. We train the model for 100 iterations with a learning rate of 0.005. We report in Table 3 the final test accuracy for different activation functions on a 5-ways-1-shots task and a 5-ways-5-shots task, respectively. The results show that AReLU shows clear advantage compared to the alternative activation functions. One noteworthy phenomenon is the performance of PAU (Molina et al., 2019): It performs well in other evaluations but not on meta learning which is probably due to its overfitting-prone nature.

## 4.6 The generalized effect of ELSA

In this experiemnt, we show that ELSA (Element-wise Sign-based Attention) can serve as a **general module** which can be plugged in to any existing activation function and obtain a performance boost for most cases.

We define a new activation $\mathcal{F}'$ the same as Eq. (7), but replace the ReLU function $\mathcal{R}$ with specified activation function. We keep the same experiment settings as Sec. 4.2. As shown in Tab: 4, after plugging with a ELSA module, we can obtain a performance boost for most cases compared with Table 1, indicating the well generalized effect of ELSA.

## 5 Conclusion

We have presented AReLU, a new learnable activation function formulated with element-wise sign-based attention mechanism. Networks implemented with AReLU can better mitigate the gradient vanishing issue and converge faster with small learning rates. This makes it especially useful in transfer learning where a pretrained model needs to be finetuned in the target domain with a small learning rate. AReLU can significantly boost the performance of most mainstream network architectures with only two extra learnable parameters per layer introduced. In the future, we would like to investigate the application/extension of AReLU to more diverse tasks such as object detection, language translation and even structural feature learning with graph neural networks.

Table 4: Mean testing accuracy (%) on MNIST for five trainings of MNIST-Conv after the *first epoch* with different optimizers and learning rates. For each activation function and each learning rate, we show results training with ELSA module. The numbers showing ELSA module improves over the original activation function (shown in Table 1) are highlighted with underline.

| Learning Rate | $1 \times 10^{-2}$ | | $1 \times 10^{-3}$ | | $1 \times 10^{-4}$ | | $1 \times 10^{-5}$ | |
|---|---|---|---|---|---|---|---|---|
| Optimizer | Adam | SGD | Adam | SGD | Adam | SGD | Adam | SGD |
| CELU (2017) | 97.81 | 97.40 | 97.06 | 93.43 | 89.90 | 65.53 | 45.99 | 13.86 |
| ELU (2015) | 97.72 | 97.51 | 96.88 | 93.38 | 89.73 | 57.85 | 40.77 | 12.96 |
| GELU (2016) | 97.99 | 97.49 | 96.93 | 92.93 | 89.83 | 42.01 | 37.40 | 10.22 |
| LReLU (2013) | 97.95 | 97.38 | 96.88 | 93.02 | 89.86 | 51.62 | 39.17 | 12.90 |
| Maxout (2013) | 97.33 | 97.47 | 96.98 | 93.50 | 90.16 | 66.74 | 48.22 | 16.37 |
| ReLU (2010) | 98.13 | 97.43 | 97.00 | 92.54 | 89.99 | 50.00 | 46.18 | 11.87 |
| ReLU6 (2010) | 97.89 | 97.60 | 97.05 | 92.83 | 89.89 | 45.15 | 39.17 | 13.40 |
| RReLU (2015a) | 97.77 | 97.37 | 97.29 | 92.75 | 89.72 | 56.28 | 37.64 | 12.61 |
| SELU (2017) | 97.37 | 97.32 | 96.81 | 93.90 | 89.91 | 68.84 | 46.14 | 11.79 |
| Sigmoid | 96.87 | 96.06 | 95.69 | 81.99 | 81.90 | 24.03 | 22.77 | 9.81 |
| Softplus (2011) | 96.80 | 97.11 | 96.25 | 91.11 | 85.68 | 39.44 | 23.92 | 12.55 |
| Swish (2017) | 97.72 | 97.45 | 96.62 | 92.47 | 89.74 | 61.69 | 38.90 | 13.18 |
| Tanh | 97.62 | 97.24 | 96.97 | 91.03 | 88.89 | 56.39 | 44.05 | 11.86 |
| APL (2014) | 98.10 | 97.48 | 96.81 | 93.12 | 89.68 | 46.81 | 28.68 | 11.37 |
| Comb (2018) | 97.95 | 97.46 | 97.08 | 93.07 | 89.43 | 51.74 | 38.97 | 13.48 |
| PReLU (2015) | 97.82 | 97.45 | 96.91 | 93.06 | 90.41 | 55.41 | 43.42 | 11.96 |
| SLAF (2019) | 95.36 | 96.74 | 96.28 | 93.80 | 85.31 | 64.03 | 22.26 | 21.06 |

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
