# OpenReview forum: "ARELU: ATTENTION-BASED RECTIFIED LINEAR UNIT"
_ICLR.cc/2021/Conference — Reject_

### Official Review · AnonReviewer1 · 2020-10-27
**Interesting work on Learnable data-adaptive activation function (AReLU)**

**Rating:** 7
**Confidence:** 3

**Review:**

Description:

This work presents a novel learned activation function called Attention-based Rectified Linear Unit (AReLU). Element-wise attention module is developed that learns a sign-based attention (ELSA) which is the novel component of AReLU towards mitigating the gradient vanishing issue. Extensive experiments and analyses have been provided on MNIST and CIFAR100 datasets. Compared with other relevant activation functions, AReLU achieves faster convergence under small learning rate because of the amplification of positive elements and suppression of negative ones with two learnable data-adaptive parameters.

Strengths:
- 13 learnable and 5 non-learnable activation function compared with the proposed AReLU. Outperforms most in convergence speed and final classification accuracy on MNIST.
- More effective training with a small learning rate.
- Impressive results for transfer learning and meta-learning
- Grad-CAM on CIFAR100 show semantically more meaningful activations

Weaknesses:
- Results are reported on CIFAR100 and MNIST datasets. The results would be more compelling if provided on datasets such as ImageNet.
- Implementation details are not provided. It would be really important to confirm if the same setup is used when comparing different activations (seed, initialization, hardware, environment)
- It is not intuitive why two different families of models were used for different datasets - MNIST-Conv (specially designed) for MNIST while Resnet-18 and VGG11 for CIFAR100

Questions:
- Could the authors confirm if the code would be made publicly available for reproducibility?
- It seems like a faster convergence and more improvement observed when SGD (compared to ADAM) is used as the optimizer (Table 1). Was a similar graph observed when using ADAM as the optim for Figure 3?
- Continuation of the above point, it would be interesting to see a similar table for CIFAR100 dataset when using ADAM / SGD as the optimizer.
- Could the authors clarify the motivation for two different learnable parameters for positive and negative elements compared to only using a single parameter? Also why sigmoid is used for positive elements.
- Have the authors considered other viable techniques to combine ELSA with ReLU (apart from addition in Section 3.3)?

Suggestions/Comments:

- Please color code (red) the learnable activation function in Table1
- Section 3.1 input a data -> input data
- Figure 1 An visualization -> Visualization
- Please correct the equation in Section 3.4 (better if equations are numbered)
- It would be helpful to provide the motivation for clamp function also in Section 3.2 apart from 3.4
- It would be helpful to be consistent with the nomenclature - MNIST Conv vs ConvMNIST in A1.
- Please confirm the dataset in A3 - CIFAR10 or CIFAR100
- It would help to explain “detaching the gradient back-propagation” in Section 3.4, independent of the framework used.

The paper is clearly written in most parts, It would be interesting to see other comments and discussions on this paper.

---------------------------------------------------------------------------------------------------------------------------------------------------------------------

Post Rebuttal update:

I would like to thank the authors for providing relevant details and a thorough rebuttal to all the issues raised by the fellow reviewers. Original rating is maintained.

---

> ### Author Response · Authors · 2020-11-14
> **Response to AnonReviewer1**
>
> Thank you for the valuable comments.
>
>
> **Q1:** Implementation details are not provided. It would be really important to confirm if the same setup is used when comparing different activations (seed, initialization, hardware, environment)
>
> **A1:** In the paper, we have provided detailed training settings for all experiments. All evaluations were conducted with the same hardware and environment. We will clarify this in the revision.
>
> -------------------
>
> **Q2:** It is not intuitive why two different families of models were used for different datasets - MNIST-Conv (specially designed) for MNIST while Resnet-18 and VGG11 for CIFAR100
>
> **A2:** Since MNIST is a relatively easy task, we designed a simple network, to better reflect the performance of activation functions. CIFAR100 is harder. We tested using several mainstream backbones.
>
> -------------------
>
> **Q3:** Could the authors confirm if the code would be made publicly available for reproducibility?
>
> **A3:** Definitely. In fact, our code has been released publicly.
>
> -------------------
>
> **Q4:** It seems like a faster convergence and more improvement observed when SGD (compared to ADAM) is used as the optimizer (Table 1). Was a similar graph observed when using ADAM as the optim for Figure 3? It would be interesting to see a similar table for CIFAR100 dataset when using ADAM / SGD as the optimizer.
>
> **A4:** In the supplemental material (Fig 2 and 3), we had provided the plots for ADAM. The results demonstrate that AReLU achieves faster convergence with either ADAM or SGD. However, the advantage with ADAM is not as significant as with SGD since ADAM adopts adaptively adjusted learning rates.
>
> -------------------
>
> **Q5:** Could the authors clarify the motivation for two different learnable parameters for positive and negative elements compared to only using a single parameter? Also why sigmoid is used for positive elements.
>
> **A5:** The derivation of AReLU is motivated by element-wise sign-based attention. The resulting two-parameter form is meaningful as it realizes the amplification of positive elements and the suppression of negative ones with independent degrees of freedom. Using sigmoid helps to stabilize the learning of $\beta$ during training (see Fig 2(b)).
>
> -------------------
>
> **Q6:** Have the authors considered other viable techniques to combine ELSA with ReLU (apart from addition in Section 3.3)?
>
> **A6:** Good question. With the current formulation, the attention module essentially learns an element-wise residual for the activated elements w.r.t. ReLU. We believe that there are alternative ways of combining ELSA with ReLU and even with other activation functions. In fact, we found in our recent experiements that ELSA can serve as a general module which can be plugged into any existing activation function and obtain a performance boost for most cases. Please see our post to all reviewers on "**The generalized effect of ELSA**".

---

### Official Review · AnonReviewer4 · 2020-10-29
**The tuning of the baseline methods are required for fair evaluations**

**Rating:** 3
**Confidence:** 4

**Review:**

Summary
This paper proposed an attention module in which each element of features is scaled based on the sign of its value.  The authors argue that the combination of this attention module and ReLU can be considered as an activation function.
This novel activation function amplifies its gradient comparing with ReLU.
The experimental results show the proposed method outperforms conventional activation functions in few-shot settings of transfer/meta learning.

Strong points
* The proposed method consistently improves the accuracy of transfer/meta learning tasks.
Weak points
* The technical novelty is limited since the derived activation function is an expansion of PReLU:
  * PReLU: min(0, a * x_i) + max(0, x_i)
  * Proposed: min(0, alpha * x_i) +  max(0, (1+sigmoid(beta))*x_i) where 0.01 < alpha < 0.99.
* Although the very specific form of ELSA is used in Section 3.2, there is no explanation of why C and sigma are introduced for ELSA.
The learning rate is not tuned for each activation function in Section 4.4 and 4.5.

Decision reason
Since the learning rates of the baseline methods seem to be not optimized, the experimental results are not convincing.
I recommend resubmitting the paper after the tuning of baseline methods.

Questions
* Why did you use momentum SGD for alpha and beta instead of using Adam or SGD as other parameters?  Their learning rates are also different from these of other parameters?
* How is the learning rate determined for the experiments in Section 4.4 and 4.5?   Even if the learning rate is tuned for baseline methods, the proposed method outperforms them?

Additional Feedback
* Although it is interesting to formulate activation functions with an element-wise attention mechanism, the merit of this new perspective is not clear for me.
* It might be easier to understand the proposed method as an expansion of PReLU since the proposed attention mechanism is much different from the attention mechanism in the community.

---

> ### Author Response · Authors · 2020-11-13
> **Response to AnonReviewer4**
>
> Thank you for the valuable comments.
>
> **Q1:** The technical novelty is limited since the derived activation function is an expansion of PReLU.
>
> **A1:** We are afraid we cannot agree. First of all, AReLU is not derived or extended from PReLU. Instead, it is motivated from a new perspective of learnable activation functions which formulates them with element-wise attention. In fact, even PReLU is clearly a learnable extension to ReLU, it is still recognized as a significant contribution.
>
> --------------------------------------
>
> **Q2:** Although the very specific form of ELSA is used in Section 3.2, there is no explanation of why C and sigma are introduced for ELSA.
>
> **A2:** $C$ and $\sigma$ are both used to confine the attention value into $[0,1]$. The use of C for the negative part not only reduces computation but also keeps consistent in form with traditional activation functions. The choice of sigmoid follows the formulation of most existing attention mechanism. Through experiments, we found that sigmoid helps stabilize the learning of $\beta$.
>
> --------------------------------------
>
> **Q3:** Why did you use momentum SGD for alpha and beta instead of using Adam or SGD as other parameters? Their learning rates are also different from these of other parameters?
>
> **A3:** This is a misunderstanding. Alpha and beta, together with all other parameters, are optimized jointly using the same optimizer and the same learning rate. Momentum update is used for alpha and beta both in ADAM and SGD. We will clarify this in the revision.
>
> --------------------------------------
>
> **Q4:** How is the learning rate determined for the experiments in Section 4.4 and 4.5? Even if the learning rate is tuned for baseline methods, the proposed method outperforms them?
>
> **A4:** Note that the learning rates use in Section 4.4 and Section 4.5 were all common choices. The configuration was not specifically tuned for a specific activation function including ours. We ensured the networks were sufficiently trained to converge for all activation functions. In Section 4.4, $1\times 10^{-5}$ is commonly used for finetuning a complicated network in transfer learning. In the meta-learning task of Section 4.5, the learning rate $0.005$ is larger than the $0.001$ used in the MAML framework (Finn et al. 2017). In fact, a smaller learning rate is more preferable to our method.
>
> --------------------------------------
>
> **Q5:** The proposed attention mechanism is much different from the attention mechanism in the community.
>
> **A5:** As a matter of fact, the element-wise attention mechanism has been practiced in the community, e.g., YOLOv4 (Bochkovskiy et al., 2020).

---

### Official Review · AnonReviewer3 · 2020-10-29
**Borderline Reject**

**Rating:** 5
**Confidence:** 5

**Review:**

Pros:
This paper is well written and is easy to follow. It focuses on the basic component in neural network, i.e. neural activation function.  Extensive experiments have been conducted and the corresponding experimental results are provided.

Cons:
While, in my view, the novelty of this paper is limited and needs more improvements. By changing the actual function performed on the input tensor X has been explored too much, such as the recent proposed FRELU: Funnel Activation for Visual Recognition. What's the difference between this paper and the proposed one? It seems the difference is limited, and they belong to the same family.

---

> ### Author Response · Authors · 2020-11-13
> **We cannot agree**
>
> With all due respect, we cannot agree with this reviewer that the novelty of a paper is judged only by the fact that the direction being studied is much explored.
>
> The FRELU paper is a nice work. However, AReLU is completely different from FRELU. *First*, AReLU is motivated by introducing the attention mechanism to the ReLU function, which we believe is a novel view to activation function. *Second*, FRELU is an extension of ReLU and PReLU by introducing a 2D spatial condition. *Third*, FRELU contains a $3\times 3$ convolution while AReLU does not involve any convolution operation. Consequently, FRELU introduces much more parameters than AReLU. *Last*, AReLU converges fast for a lower learning rate which benefits transfer learning.
>
> This reviewer says "It seems the difference is limited, and they belong to the same family."
>
> This arguement is quite vague and we are unsure what the reviewer means by "the same family". They are both activation functions but they are completely different contributions.

---

### Official Review · AnonReviewer2 · 2020-10-31
**A new type of ReLU unit inspired by attention mechanism**

**Rating:** 6
**Confidence:** 3

**Review:**

[Overview]

In this paper, the authors proposed a new activation function called AReLU which introduces an attention mechanism to the original ReLU function. Based on this new activation function, the output will be adaptively adjusted by the two learnable parameters \alpha and \beta. This kind of adaptive adjustment can be thought of as an attention mechanism undertaken over each element in the input feature map. It will in general amplify the positive elements while suppressing the negative ones, and the parameters \alpha and \beta will be adjusted adaptively based on the activation values. The experimental results showed that AReLU can achieve much better performance with small learning rates while comparable performance with fairly large learning rates. This inspires another set of transfer learning experiments that demonstrate the effectiveness of AReLU.

[Strength]

1. This paper proposed a simple element-aware activation function. It is built based on ReLU but differs from ReLU in that it has two learnable parameters and will adaptively augment the positive inputs while suppressing the negative ones.

2. The mechanism behind the proposed AReLU seems reasonable. It increases the scale to later than one for positive responses while decreasing the scale for negative responses.

3. The experiments showed that AReLU is more robust to the learning rates compared with other activation functions, either non-learnable or learnable ones. Also, for a lower learning rate, the convergence speed is much faster than other activation functions. This property facilitates the transfer learning scenarios as shown in Sec. 4.4.

[Weakness]

1. It seems that AReLU will hurt the performance when the learning rate is reasonably set, as shown in the first super-column (1e-2) in Table 1. Also, it is not clear about the final performance on CIFAR-100. Overall, it hard for me to determine whether the proposed AReLU can have generic benefits to the training under different settings.

2. In Fig.2 (b),  the values of \alpha are different for different learning rates, while (1+\sigma(\beta)) converges to similar value. This raises the question that whether it is necessary to set a learnable \beta or not because we can simply fix the value of \beta at the beginning. Even for \alpha, it is not clear whether it is necessary to learn it because the authors did not show the final accuracies for different settings.

3. In Table 3, the authors showed that AReLU is more suitable for transfer learning with a low learning rate. However, the comparison is a bit unfair, for other activation functions, we can simply increase the learning rate and it might be the case that they can achieve better performance with a larger learning rate. Another baseline is, as pointed above, we can simply set \alpha less than 1 (say 0.5) while 1+\sigma(\beta) to around 1.5.

4. The proposed AReLU is good for learning the parameters in an adaptive manner. However, does this will introduce another problem, i.e., overfitting? With less amount of data, this adaptiveness may hurt the generalization ability of the learned network. In this paper, the authors did not study this aspect.

5. If the goal of AReLU is mainly to address the vanishing gradient, then what if we remove the sigmoid to have a (1+\beta)? What will be the outcome in this case? I would like to hear from the authors about this.

[Summary]

Overall I think this paper is well-written and with a fluent flow to follow. The intuition behind the proposed AReLU is clear to understand and I like it. Experimental results demonstrated the effectiveness of AReLU, especially under training+finetuning settings. However, as pointed above, I am still concerned about the generalization of AReLU and the necessity of applying AReLU in the training. I would like to hear more from the authors in the rebuttal.

---

> ### Author Response · Authors · 2020-11-13
> **Response to AnonReviewer2**
>
> Thank you for the valuable comments.
>
> **Q1**: It seems that AReLU will hurt the performance when the learning rate is reasonably set, as shown in the first super-column (1e-2) in Table 1. Also, it is not clear about the final performance on CIFAR-100. Overall, it hard for me to determine whether the proposed AReLU can have generic benefits to the training under different settings.
>
> **A1**: From Table 1, AReLU improves over the baseline activation function ReLU by a large margin for different learning rates. It achieves comparable accuracy with the state-of-the-art activation functions (CELU, PAU, etc.). Therefore, AReLU is definitely not "hurting" the performance. The final performance on CIFAR-100 is given in the supplemental material (Table 4), where we show that AReLU achieves SOTA performance.
>
> --------------------------------
>
> **Q2**: In Fig 2(b), the values of $\alpha$ are different for different learning rates, while $(1+\sigma(\beta))$ converges to similar value. This raises the question that whether it is necessary to set a learnable $\beta$ or not because we can simply fix the value of $\beta$ at the beginning. Even for $\alpha$, it is not clear whether it is necessary to learn it because the authors did not show the final accuracies for different settings.
>
> **A2**: First of all, we found that using a fixed $\beta$ causes a slower training convergence. In Fig 2, the reason why the final values of $\beta$ are close is that the network here is relatively simple. In a more complicated network, the final values of $\beta$ are not necessarily close. We did show the final accuracies for different settings in the supplemental material (Fig 2 and 3).
>
> --------------------------------
>
> **Q3**: In Table 3, the authors showed that AReLU is more suitable for transfer learning with a low learning rate. However, the comparison is a bit unfair, for other activation functions, we can simply increase the learning rate and it might be the case that they can achieve better performance with a larger learning rate. Another baseline is, as pointed above, we can simply set $\alpha$ less than 1 (say 0.5) while $1+\sigma(\beta)$ to around 1.5.
>
> **A3**: Note that we uniformly used a large learning rate for all methods being compared and made sure their training all converge. About the baseline pointed by the reviewer, we did a test and below is the results:
>
> |  lr of SGD |  $\quad1\times 10^{-2}$ |  $\quad1\times 10^{-3}$ |  $\quad1\times  10^{-4}$ |  $\quad1\times 10^{-5}$ |
> | :------------- | :----------: |:-----------: | :-----------: | :-----------: |
> |  AReLU     |      97.3    |      93.1    |      47.8    |      14.3    |
> |  Baseline  |      96.9    |      89.9    |     20.0     |      9.24    |
>
> |  lr of ADAM |  $\quad1\times 10^{-2}$ |  $\quad1\times 10^{-3}$ |  $\quad1\times 10^{-4}$ | $\quad 1\times 10^{-5}$ |
> | :------------- | :----------: |:-----------: | :-----------:| :-----------: |
> |     AReLU      |      98.0    |      97.1    |      90.4    |      38.4    |
> |     Baseline   |      97.7    |      96.6    |     87.7     |      27.9    |
>
> --------------------------------
>
> **Q4**: The proposed AReLU is good for learning the parameters in an adaptive manner. However, does this will introduce another problem, i.e., overfitting? With less amount of data, this adaptiveness may hurt the generalization ability of the learned network. In this paper, the authors did not study this aspect.
>
> **A4**: We have discussed in Sec 4.5 and shown in Fig 4 about the overfitting issue. AReLU introduces only one extra parameter compared to PReLU, and we did not see an overfitting problem in our experiments. We also compared to PAU which has the overfitting issue.
>
> --------------------------------
>
> **Q5**: If the goal of AReLU is mainly to address the vanishing gradient, then what if we remove the sigmoid to have a $1+\beta$? What will be the outcome in this case? I would like to hear from the authors about this.
>
> **A5**: Adding the sigmoid better confines the positive values. We had experimented $1+\beta$ and found that it always leads to infinitely large positive activations and makes the network untrainable. Moreover, sigmoid also helps stabilize the $\beta$ during training (see Fig 2(b)).

---

### Author Response · Authors · 2020-11-14
**Thanks and a brief summary of responses**

Dear Reviewers,

Thank you for the valuable comments and constructive suggestions.

We are encouraged to see a wide appreciation of the novel view of activation function and the merit of AReLU in accelerating training with small learning rates benefiting important applications like transfer learning and meta-learning.

AnonReviewer4 questioned the technical novelty by commenting that AReLU is an extension of PReLU. We believe this is a misunderstanding. Actually, AReLU is not derived or extended from PReLU. Instead, it is motivated from a new perspective of learnable activation functions which formulates them with element-wise attention. Other misunderstandings or clarity issues have been addressed in the detailed response.

AnonReviewer3 claims that the paper has limited novelty based only on the fact that the direction being studied is "much explored". We cannot agree with this argument. We also address how our method is completely different from the recent FRELU work pointed by the reviewer.

We have provided detailed response to each reviewer and we are ready to submit a revision with all review comments/suggestions addressed and to enhance the supplemental material. By doing so, we hope that the reviewer's concerns could be largely addressed.

Thank you,

The authors.

---

### Author Response · Authors · 2020-11-14
**The generalized effect of ELSA (Element-wise Sign-based Attention)**

In this post, we'd like to provide some notable results we've obtained recently. We show that ELSA (Element-wise Sign-based Attention) can serve as a **general module** which can be plugged into any existing activation function and obtain a performance boost for most cases.

Suppose $F$ is some activation function and $Elsa$ is the ELSA function (see its definition in Section 3.3 of the main paper), the new activation function is defined as: $F^{\prime}(x) = F(x) + Elsa(x)$.

In the table below, we report the mean testing accuracy (%) of MNIST-Conv trained with different activation functions. For each activation function and each learning rate, we show results of both ***with*** and ***without*** ELSA module. The numbers showing ELSA module (**/w**) improves over the original activation function (**w/o**) are highlighted with **boldface**. The results in this table were obtained with the ADAM optimizer; SGD results can be found in the post following this one.

|    **LR**    | $\quad10 ^ {-2}\quad$ | $\quad10 ^ {-2}\quad$ | $\quad10 ^ {-3}\quad$ | $\quad10 ^ {-3}\quad$ | $\quad10 ^ {-4}\quad$ | $\quad10 ^ {-4}\quad$ | $\quad10 ^ {-5}\quad$ | $\quad10 ^ {-5}\quad$ |
| :------ | :---------: | :---------: | :---------: | :---------: | :---------: | :---------: | :---------: | :---------: |
| **ELSA**  |  **w/o**   |   **w/**   |  **w/o**   |   **w/**   | **w/o**   |   **w/**   |  **w/o**   |  **w/**   |
|   APL    |     97      |  **98.1**   |    94.67    |  **96.81**  |    76.73    |  **89.68**  |    13.28    |  **28.68**  |
|   GELU   |  98.49 |    97.99    |    95.79    |  **96.93**  |    83.72    |  **89.83**  |    15.20    |  **37.40**  |
|  Maxout  |    97.04    |  **97.33**  |    96.14    |  **96.98**  |    84.81    |  **90.16**  |    18.83    |  **48.22**  |
|   Comb   |  98.28  |    97.95    |    95.79    |  **97.08**  |    83.91    |  **89.43**  |    20.22    |  **38.97**  |
|   SLAF   |  96.3   |    95.36    |    95.32    |  **96.28**  |    72.67    |  **85.31**  |    10.04    |  **22.26**  |
|  Swish   |  98.1   |    97.72    |    95.91    |  **96.62**  |    83.91    |  **89.74**  |    11.39    |  **38.90**  |
|   ReLU   |    97.75    |  **98.13**  |    95.4     |  **97.00**  |    84.02    |  **89.99**  |    15.25    |  **46.18**  |
|  ReLU6   |    97.77    |  **97.89**  |    96.09    |  **97.05**  |    81.39    |  **89.89**  |    14.33    |  **39.79**  |
| Sigmoid  |    47.16    |  **96.87**  |    83.59    |  **95.69**  |    11.37    |  **81.90**  |    10.52    |  **22.77**  |
|  LReLU   |    97.8     |  **97.95**  |    95.86    |  **96.88**  |    84.08    |  **89.86**  |    15.41    |  **39.17**  |
|   ELU    |  97.82  |    97.72    |    96.22    |  **96.88**  |    85.67    |  **89.73**  |    19.77    |  **40.77**  |
|  PReLU   |  98.22  |    97.82    |    95.65    |  **96.91**  |    85.81    |  **90.41**  |    14.51    |  **43.42**  |
|   SELU   |    97.25    |  **97.37**  |    96.61    |  **96.81**  |    85.36    |  **89.91**  |    30.04    |  **46.14**  |
|   Tanh   |    96.93    |  **97.62**  |    96.45    |  **96.97**  |    79.25    |  **88.89**  |    27.05    |  **44.05**  |
|  RReLU   |  98.09  |    97.77    |    95.65    |  **97.29**  |    84.51    |  **89.72**  |    16.53    |  **37.64**  |
|   CELU   |    97.76    |  **97.81**  |    96.21    |  **97.06**  |    84.01    |  **89.90**  |    24.84    |  **45.99**  |
| Softplus |    96.38    |  **96.8**   |    93.83    |  **96.25**  |    51.83    |  **85.68**  |    10.21    |  **23.92**  |

---

> ### Author Response · Authors · 2020-11-14
> **The generalized effect of ELSA: Results by SGD**
>
> In the table below, we report the mean testing accuracy (%) of MNIST-Conv trained with different activation functions. For each activation function and each learning rate, we show results of both with and without ELSA module. The numbers showing ELSA module (**/w**) improves over the original activation function (**w/o**) are highlighted with boldface. The results in this table were obtained with the SGD optimizer.
>
> |    **LR**    | $\quad10 ^ {-2}\quad$ | $\quad10 ^ {-2}\quad$ | $\quad10 ^ {-3}\quad$ | $\quad10 ^ {-3}\quad$ | $\quad10 ^ {-4}\quad$ | $\quad10 ^ {-4}\quad$ | $\quad10 ^ {-5}\quad$ | $\quad10 ^ {-5}\quad$ |
> | :------ | :---------: | :---------: | :---------: | :---------: | :---------: | :---------: | :---------: | :---------: |
> | **ELSA**  |  **w/o**   |   **w/**   |  **w/o**   |   **w/**   | **w/o**   |   **w/**   |  **w/o**   |  **w/**   |
> |   APL    |    95.71    |  **97.48**  |    17.81    |  **93.12**  |    9.39     |  **46.81**  |  11.83  |    11.37    |
> |   GELU   |    94.9     |  **97.49**  |    12.55    |  **92.93**  |    11.49    |  **42.01**  |    9.60     |  **10.22**  |
> |  Maxout  |    95.81    |  **97.47**  |    71.75    |  **93.5**   |    10.79    |  **66.74**  |    9.06     |  **16.37**  |
> |   Comb   |    95.97    |  **97.46**  |    35.95    |  **93.07**  |    10.59    |  **51.74**  |    10.18    |  **13.48**  |
> |   SLAF   |  97.07  |    96.74    |    83.35    |  **93.80**  |    14.12    |  **64.03**  |    11.32    |  **21.06**  |
> |  Swish   |    94.02    |  **97.45**  |    11.44    |  **92.47**  |    10.69    |  **61.69**  |    9.47     |  **13.18**  |
> |   ReLU   |    95.02    |  **97.43**  |    36.01    |  **92.54**  |    10.68    |  **50.00**  |    8.73     |  **11.87**  |
> |  ReLU6   |    95.32    |  **97.60**  |    43.42    |  **92.83**  |    10.23    |  **45.15**  |    9.56     |  **13.40**  |
> | Sigmoid  |    11.04    |  **96.06**  |    11.35    |  **81.99**  |    9.92     |  **24.03**  |  10.1   |    9.81     |
> |  LReLU   |    95.59    |  **97.38**  |    35.9     |  **93.02**  |    10.28    |  **51.62**  |    10.73    |  **12.90**  |
> |  PReLU   |    95.72    |  **97.45**  |    45.73    |  **93.06**  |    12.08    |  **55.41**  |    9.88     |  **11.96**  |
> |   SELU   |    96.52    |  **97.32**  |    53.33    |  **93.90**  |    16.49    |  **68.84**  |    9.59     |  **11.79**  |
> |   Tanh   |    94.22    |  **97.24**  |    57.7     |  **91.03**  |    11.73    |  **56.39**  |    10.31    |  **11.86**  |
> |  RReLU   |    95.88    |  **97.37**  |    53.33    |  **92.75**  |    9.57     |  **56.28**  |    10.28    |  **12.61**  |
> |   CELU   |    96.12    |  **97.40**  |    62.81    |  **93.43**  |    13.07    |  **65.53**  |     9.6     |  **13.86**  |
> | Softplus |    90.90    |  **97.11**  |    11.14    |  **91.11**  |    9.19     |  **39.44**  |    9.89     |  **12.55**  |
> |   ELU    |    96.17    |  **97.51**  |    58.1     |  **93.38**  |    14.07    |  **57.85**  |    10.13    |  **12.96**  |

---

### Decision · Program_Chairs · 2021-01-07
**Final Decision**

**Decision:**

Reject

**Comment:**

I would like to thank the authors for the their time and effort on this work. The paper is proposing an activation function that combines RELU like piecewise activation functions and a primitive attention mechanism. Then, they show that their proposed method works better in transfer settings.

I think the approach authors taking here is more akin to a gating mechanism rather than an attention. So I would recommend the authors to change the name perhaps Gated Rectified Linear Units. The paper is interesting, but I agree with AnonReviewer4, that the experiments are not very convincing focusing on small scaled experiments in the supervised learning setting only. I would recommend the authors to compare their approach against other results from the literature as well. As it is right now it is not clear how significant the results in this paper are. I don't think the transfer and meta-learning experiments are very well-motivated in this paper. I would recommend the authors to better motivate those results.

After considering my suggestions above and the comments from the reviewers I would recommend the authors to consider resubmitting to another conference.